# ADAPTIVE MIXTURE OF LOW-RANK FACTORIZATIONS FOR COMPACT NEURAL MODELING

## ABSTRACT

Modern deep neural networks have a large amount of weights, which make them difficult to deploy on computation constrained devices such as mobile phones. One common approach to reduce the model size and computational cost is to use low-rank factorization to approximate a weight matrix. However, performing standard low-rank factorization with a small rank can hurt the model expressiveness and significantly decrease the performance. In this work, we propose to use a mixture of multiple low-rank factorizations to model a large weight matrix, and the mixture coefficients are computed dynamically depending on its input. We demonstrate the effectiveness of the proposed approach on both language modeling and image classification tasks. Experiments show that our method not only improves the computation efficiency but also maintains (sometimes outperforms) its accuracy compared with the full-rank counterparts.

## 1 INTRODUCTION

Modern neural networks usually contain millions of parameters (Krizhevsky et al., 2012; Simonyan & Zisserman, 2014), and they are difficult to be deployed on mobile devices with limited computation resources. To solve this problem, model compression techniques are proposed in recent years. For example, (Wu et al., 2018; Li et al., 2016; Han et al., 2015a) try to limit the weights (and activations) of neural networks to lower precisions by quantization. This can save the model size by 4 to 16 times. While quantization can reduce the number of bits per weight, it cannot reduce the number of weights. To reduce this redundancy, (Han et al., 2015b; Ullrich et al., 2017; Louizos et al., 2017) propose pruning the weight matrices, leading to sparse neural networks that require less computation. However, sparse neural networks often require specialized ASIC or FPGA to accelerate (Han et al., 2016; 2017).

Alternatively, low-rank factorization is a popular way of reducing the matrix size. It has been extensively explored in the literature (Lu et al., 2016; Nakkiran et al., 2015; Jaderberg et al., 2014; Yu et al., 2017). Mathematically, a large weight matrix $W \in \mathbb{R}^{m \times n}$ is factorized to two small rank-$d$ matrices $U \in \mathbb{R}^{m \times d}$, $V \in \mathbb{R}^{n \times d}$ with $W = UV^T$. Since both $U$ and $V$ are dense, no sparsity support is required from specialized hardware. It naturally fits the general-purpose, off-the-shelf CPUs and GPUs.

To significantly reduce the model size and computation, the rank $d$ in the low-rank factorization needs to be small. However, a small rank can limit the expressiveness of the model (Yang et al., 2018) and lead to worse performance. To understand the limitations, given a $n$-dim feature vector $h$, we observe that $V^T h$, as in $U(V^T h)$, is a linear projection from a high-dimensional space ($n$ dims) to a low-dimensional space ($d$ dims). This can lead to a significant loss of information. The conflict between the rank $d$ and the model expressiveness prevents us from obtaining a both *compact* and *accurate* model.

To address the dilemma, we propose to increase the expressiveness by learning an adaptive, input-dependent factorization, rather than performing a fixed factorization of a weight matrix. To do so, we use a mixture of multiple low-rank factorizations. The mixing weights are computed based on the input. This creates an adaptive linear projection from a high-dimensional space to a low-dimensional space. Compared to the conventional low-rank factorization, the proposed approach can significantly improve its performance while only introducing a small additional cost.

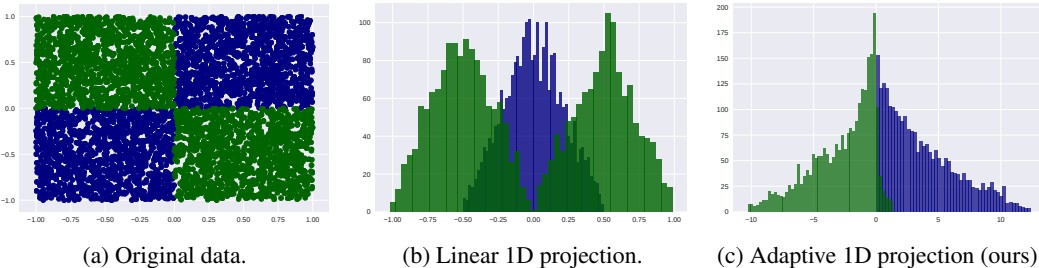

| (a) Original data. | (b) Linear 1D projection. | (c) Adaptive 1D projection (ours). |

Figure 1: A toy classification problem with a rank-1 factorization of the weight matrices. (b) and (c) are distributions of 2D data in the 1D projected space. The linear projection to lower dimension leads to significant information loss (results in 83% classification accuracy), while our proposed approach learns to adaptively avoid this (achieving 97% classification accuracy). (c) is the distribution of projection through a random matrix followed by tanh.

To demonstrate the effectiveness of adaptive low-rank factorization, we experiment with both recurrent and convolutional neural networks on language modeling and image classification tasks. Experimental results on both tasks show that our method consistently improves upon conventional low-rank factorization. On the Penn Tree Bank dataset, we achieved 40% reduction in FLOPs, and 1.7 better perplexity than the full rank baseline LSTM for language modeling. On ImageNet dataset, we use 48% less computation, 12% less parameters, but achieve 2.5% better Top-1 accuracy than MobileNet-V1 (Howard et al., 2017). Compared to MobileNet-V2 which utilizes a standard low-rank bottleneck structure, our proposed method achieves 1.95% better Top-1 accuracy with less than 1% extra FLOPs, which is significant given that MobileNet-V2 is already very compact.

## 2 LOW-RANK FACTORIZATION AND THE LINEAR BOTTLENECK

A common linear transformation between two spaces can be represented by a linear function $\mathcal{F}$ : $\mathbb{R}^n \to \mathbb{R}^m$, $\mathcal{F}(h; W) = Wh$ where $W \in \mathbb{R}^{m \times n}$ and $h \in \mathbb{R}^n$. To reduce the size and computation of this linear transformation, a low-rank factorization of $W$, i.e. $W = UV^\top$, can be applied, where $U \in \mathbb{R}^{m \times d}$, $V \in \mathbb{R}^{n \times d}$, and $d < \min(m, n)$. With this factorization, we can compute the transformation with $h' = UV^\top h$. This reduces computations from $O(mn)$ to $O((m + n)d)$. In the context of neural networks where $W$ represents a weight matrix for a layer, both $U$ and $V$ can be learned using gradient-based algorithms.

From the model compression perspective, we want to minimize the rank $d$, since that relates to smaller model size and computation [1]. However, the expressiveness of the transformation is limited by the rank $d$. By applying the factorized transformation $h' = UV^\top h$ in the reverse order, i.e. $h' = U(V^\top h)$, we observe that the first transformation for $h$ is to project it from a high-dimensional to a low-dimensional space since $d < n$. The latent feature distribution in high dimensional space may be either high dimensional, or lie on a non-linear manifold. In either case, projecting it into the low-dimensional space can lead to significant information loss if $d$ is small. This is less appealing for preserving information for latter layers.

To demonstrate the expressiveness issue, we conduct a toy classification task in 2D spaces. We first generate a 2D dataset with XNOR labels (two diagonal blocks are labeled with the same class) 1a. A non-linear classifier with one hidden layer is trained to predict the output probability by $P(y|x) = \text{softmax}(W_2\sigma(W_1x))$ where $W_1 \in \mathbb{R}^{2 \times 2}$, and we factorize $W_1$ using two $2 \times 1$ matrices, i.e. $W_1 = UV^T$. Since the rank $d = 1$, the 2D data points are first projected into 1D space and then projected into class probabilities. The visualization of one-dim space is shown in Figure 1b. We observe a significant amount of previously separated data points are now overlapped, and the class information is lost. We attribute this loss of information to the *linear bottleneck*.

This limitation cannot be solved by adding non-linear activation at the bottleneck (after the linear transformation), since the information is already lost before the application of the non-linearity. Even worse, adding lossy non-linearity to a low-dimensional manifold will further negatively impact the

---

[1]The compression ratio, as well as the ratio of FLOPS, is $mn/((m + n)d)$. A smaller $d$ is better.

network's performance, as pointed out in (Howard et al., 2017). Therefore, a new approach to boost the expressiveness of the linear bottleneck without much overhead is in demand.

## 3 ADAPTIVE MIXTURE OF LOW-RANK FACTORIZATIONS

To overcome the linear bottleneck in the low-rank factorization approach presented above, we propose to use an unnormalized learned mixture of low-rank factorizations whose mixing weights are computed adaptively based on the input. More specifically, denoting the input by $h$ and the number of mixture components by $K$, we decompose a large weight matrix by

$$W(h) = \sum_{k=1}^{K} \pi_k(h) U^{(k)} \big(V^{(k)}\big)^{\top}, \tag{1}$$

where $\pi(\cdot) : \mathbb{R}^n \to \mathbb{R}^K$ is the function which maps each input to its mixture coefficients, and $U^{(k)} \in \mathbb{R}^{m \times d/K}$, $V^{(k)} \in \mathbb{R}^{n \times d/K}$. For example, $\pi$ can be a small neural network. This introduces a small amount of extra parameters and computation. We will later discuss the details of efficient ways to implement the mixture function $\pi$.

If $\pi_k$, $k = 1, ..., K$, is chosen to be constant (input independent), it can be absorbed into either $U^{(k)}$ or $V^{(k)}$. Thus, the proposed method reduces to the low-rank factorization. This is evidenced by rewriting $W$ as $W = [\pi_1 U^{(1)}, ..., \pi_K U^{(K)}][V^{(1)}, ..., V^{(K)}]^{\top}$. In other words, the conventional low-rank factorization can be considered as a special case of our method.

**Adaptive mixing weights $\pi(h)$.**  The mixing weights can encode important information that we can use to increase the expressiveness of the projected low-dimensional space. Under our framework, the generation of the mixing weights $\pi(h)$ is flexible. A straight-forward approach is to use a non-linear transformation of the input to the weight matrix. For example, $\pi(h) = \sigma(Ph)$, where $\sigma$ is a non-linear transformation, such as sigmoid or hyperbolic tangent function, and $P \in \mathbb{R}^{K \times n}$ is an extra weight matrix. This adds some extra parameters and computation to the model since the linear projection that we construct is $\mathbb{R}^n \to \mathbb{R}^K$. To further reduce the parameter and computation in the mixing weights $\pi$, we propose the following strategies.

*Pooling before projection.* We do not require the whole input to compute the mixture function $\pi$. Instead, we can apply pooling to the input $h$ before projection. For example, a global average pooling can be applied if the input is a 3D tensor (for images); for a 1D vector, we can segment the vector and average each segmentations. By applying pooling, we can both save the computation and better capture the global information.

*Random projection.* To reduce the number of parameters in the linear projection of $h$, we can use a random matrix $P_{\text{random}}$ in place of a fully adjustable $P$, i.e. $\pi(h) = \sigma(P_{\text{random}}h)$. Although we cannot control freely the information captured by a random matrix, hopefully the linear projection induced by $U$ and $V$ can adaptively learn features that are complementary to the mixture weights. Note that we can simply save a seed to recover the random matrix, but it still requires the same amount of memory and computation as the fully adjustable linear projection of $h$.

**Increased expressiveness.**  Due to the use of data-dependent mixing weights for multiple low-rank factorizations, we expect the expressiveness of the model to increase. To demonstrate this intuition, in the toy example of Figure 1, the distribution of adopting our approach that computes the mixing weights from a random matrix $P$ and tanh non-linearity is shown in Figure 1c. Compared to conventional linear bottleneck in Figure 1b, we see a better separation among data between the two classes, leading to improved classification accuracy (97% vs. 83%). This is not surprising since the mixing weights encode certain class distribution in 2D space and augments the linear projection. The original overlapped projection is now better separated.

More precisely, the adaptive mixing weights introduce a non-linear transformation into the high-to-low-dimensional projection that can be more expressive. Since each $W(h)$ is a data-dependent low-rank matrix, there is no constant linear weight independent to the input (even a full-rank matrix) that can mimic the transformation $W(h)$ induced by our proposed method. It is worth noting that generating the whole weight matrices can be very expensive. Our method can be seen as a swift

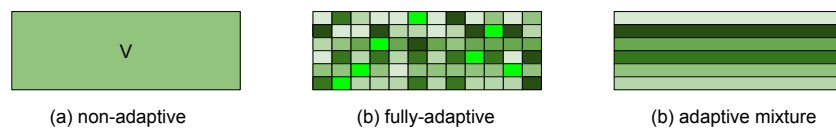

(a) non-adaptive          (b) fully-adaptive          (b) adaptive mixture

Figure 2: Illustration of different types linear projection weights $V$ colored by responses to a particular input $h$: (a) A data-independent non-adaptive weight matrix, (b) fully adaptive weight matrix which can be very expensive, (c) the proposed adaptive mixture approach.

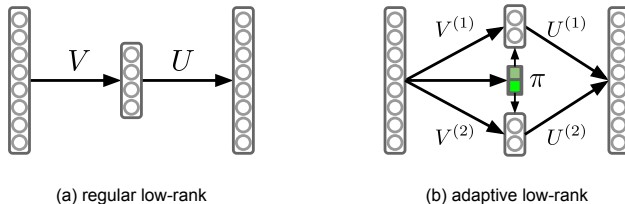

(a) regular low-rank          (b) adaptive low-rank

Figure 3: (a) regular factorization and (b) adaptive mixture of low-rank factorizations. First compute $z_k = \pi_k(h)((V^{(k)})^T h)$ and then $h' = \sum_k U^{(k)} z_k$, where $z$ can be treated as the middle layer. Techniques like pooling can be applied to compute $\pi$ so it does not induce much extra parameters and costs.

approach to generate the weights by adaptively adjusting mixing weights for the linear bottleneck. It assigns weights into groups and dynamically controls them at the group level, as demonstrated in Figure 2.

To efficiently compute the whole linear transformation $\sum_{k=1}^K \pi_k(h) U^{(k)} (V^{(k)})^\top h$, we use the reverse order, i.e. first computing the linear projection into low-dimensional space with mixing weights, i.e. $z_k = \pi_k(h)((V^{(k)})^T h)$, and then map to a higher dimensional space, i.e. $h' = \sum_k U^{(k)} z_k$. This reduces the FLOPs and also avoids to store different weight matrices $W(h)$ for different training examples in a mini-batch. An illustration of the computation framework is presented in Figure 3. Compared to original low-rank factorization, extra parameters and computation cost are from the mixing weights. They can be very small with techniques like pooling aforementioned. We also introduce a way in appendix to avoid breaking a bulk matrix multiplication to segments.

## 4 EXPERIMENTS

In this section, we first showcase the linear bottleneck in MNIST with multi-layer perceptron (MLP), and demonstrate how our proposed method improves upon the regular low-rank factorization. Then we conduct extensive experiments on both recurrent neural networks for language modeling and convolutional neural networks for image recognition on ImageNet.

### 4.1 ADAPTIVE LOW-RANK FACTORIZATION FOR MLP

In this experiment, we construct a MLP for digit recognition using MNIST dataset. We use a simple one-layer MLP of 300 hidden units (whose input and output sizes are 784 and 10, respectively), and it can be written as $P(y|x) = \text{softmax}(W_2 \sigma(W_1 x))$. We factorize $W_1 \in \mathbb{R}^{784 \times 300}$ with a rank-$d$ matrix. We use $d = 2$ in this case to better expose the issue of linear bottleneck and better visualize the latent data distribution. We also set the number of mixture $K = 2$. To compute mixing weights, we first reduce $x \in \mathbb{R}^{784}$ to $\mathbb{R}^{28}$ with a segment-based mean pooling, so that the extra parameters is of dimension $28 \times 2$ and computations only accounts for a small amount ($< 1\%$ of the overall parameters and FLOPs).

The accuracy of non-adaptive low-rank factorization is only 73%, but the adaptive version is 82.6%, a significant boost. We visualize data distributions in the 2D feature space for non-adaptive and adaptive low-rank factorizations in Figure 4 (we also present additional figures with TSNE (Maaten

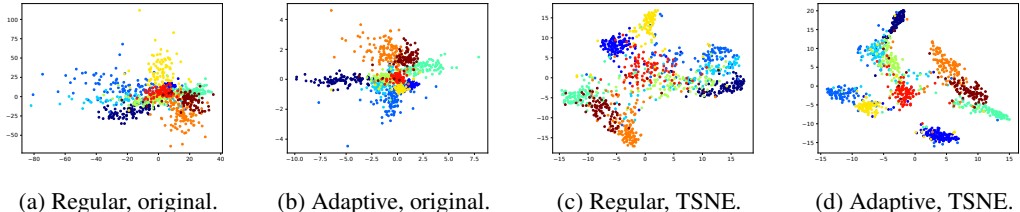

| (a) Regular, original. | (b) Adaptive, original. | (c) Regular, TSNE. | (d) Adaptive, TSNE. |

Figure 4: Visualization of low-rank projected 2D space. Non-adaptive versus adaptive low-rank, in both original 2-d space and TSNE enhanced 2D space. With adaptive mixtures, we observe better separation among data points of different classes, closely positioning of the data of the same class. (Best view in color.)

& Hinton, 2008) to enhance the visualization). We can see that with the adaptive mixing weights, data points of different classes are better separated in the projected low-dimensional space.

## 4.2 COMPRESSING RECURRENT NEURAL NETWORKS FOR LANGUAGE MODELING

Recurrent neural networks (RNNs) are widely used in language modeling, machine translation and sequence modeling in general. In RNNs, we need to compute the transition of hidden states, e.g., $h_t = \sigma(W_h h_{t-1} + W_x x_t + b)$ at each time step. The transition weight matrices can be very large and very suitable for low-rank factorizations (Lu et al., 2016).

In our experiment, we adopt the same Long Short Term Memory (LSTM) models and follow the settings from a previous state-of-the-art model (Zaremba et al., 2014) for language modeling, and use Penn Tree Bank (PTB) as well as Text8 datasets. More specifically, we use the medium sized model introduced in (Zaremba et al., 2014), which consists of two layers LSTM with 650 hidden units. Dropouts of 0.5 are added between different layers.

The performance of a language model is commonly measured with perplexity, which is basically the exponential of average negative likelihood, and a smaller number is more desirable. By default, we directly factorize a concatenated joint weight matrix in LSTM, and make comparisons using different rank-$d$, measured by the ratio to the averaged input size $n$ and output size $m$, i.e. $2d/(m+n)$. we set the number of mixtures to the rank, i.e. $K = d$, since in a good compression the rank $d$ is expected to be small. More details are presented in the supplementary materials. We use the sigmoid activation for computing the mixing weights.

Our main baseline is the regular low-rank factorization, and we test three variants of the proposed model, each with different ways of computing mixing weights, namely (1) MIX-ALL-PJ: direct linear projection of the input vector $h$, (2) MIX-POOL-PJ: linear projection after segment-based mean pooling of the input vector $h$, and (3) MIX-RND-PJ: use a random projection for the input vector $h$. Among these adaptive projection methods, MIX-ALL-PJ has a large amount of extra parameters, MIX-POOL-PJ has a small amount of extra parameters, and MIX-RND-PJ has no extra parameters. We compute the FLOPs of a single time-step of applying LSTM (excluding the output softmax layer), and the perplexity associated to different settings.

The results are shown in Figure 5. Firstly, with adaptive mixtures, the low-rank factorization model achieved 40% reduction in FLOPs, and even surpassed the performance of the full matrix baseline by decreasing the perplexity by 1.7 points in Penn Tree Bank. Secondly, as we decrease the rank and reduces the FLOPs, we observe the performance degradation, which is as expected. However, the use of adaptive mixtures can significantly improve the performance compared with regular, non-adaptive low-rank factorization (e.g. in Text8 data set, we reduce the FLOPs by 70% while maintaining the same perplexity). Thirdly, we see that different ways of generating the mixing weights have impacts on the performance, and the trade-off between performance and FLOPs/model size. We find that using pooling before projection can be a good choice for computing the mixing weights $\pi$. It not only reduces the computation and parameter size, but can better capture the global information and achieve better accuracy.

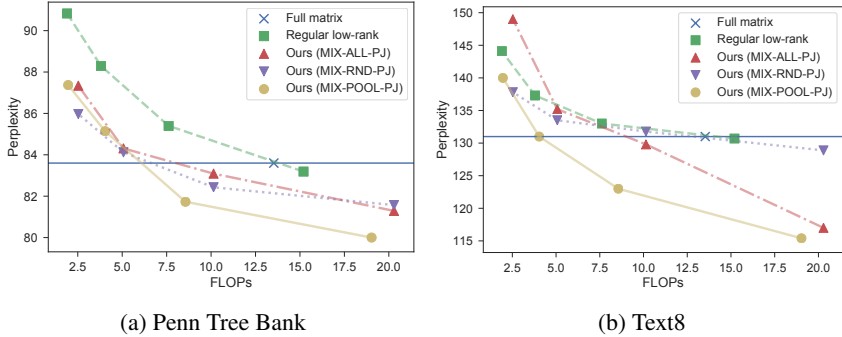

(a) Penn Tree Bank        (b) Text8

Figure 5: FLOPs vs. perplexity. The horizontal line is the full LSTM's baseline accuracy. We also compare variants of the proposed approaches with regular low-rank factorization, indicated by different colors and markers. Lower perplexity is better.

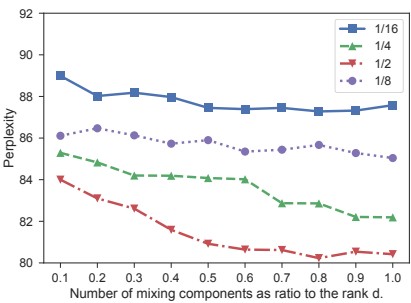

Figure 6: Perplexity vs. number of mixing components. Different curves denote for different rank-$d$, as a ratio to the averaged input and output dims, i.e. $\frac{2d}{m+n}$.

We further explore the effects of the number of mixtures used in our method by using different ratio of mixtures to the low-rank dimensionality $d$. The results are shown in Figure 6. We find that using more mixtures generally leads to better results, although the performance start to plateau when the number of mixtures is large enough. However, to obtain a larger compression rate and speedup, the rank-$d$ we use in the low-rank factorization can be already small, thus the extras of using different number of mixtures may not differ too much.

## 4.3 COMPRESSING CONVOLUTIONAL NEURAL NETWORKS FOR IMAGE RECOGNITION

We further demonstrate the effectiveness of the proposed approach on compressing CNN models. We chose to use modern compact CNN models as the baseline (which are harder to compress), rather than using the bulky CNN models (which is easier to compress). Recently, a major advance in designing compact CNNs architecture is so called *depth-wise separable convolutions* (Chollet, 2016; Howard et al., 2017; Sandler et al., 2018). Compared to a standard convolutional kernel that computes the transformation ($\mathbb{R}^{H \times W \times C} \to \mathbb{R}^{H \times W \times C'}$) [2], a depth-wise separable convolution includes a depth-wise convolution ($\mathbb{R}^{H \times W \times 1} \to \mathbb{R}^{H \times W \times 1}$) and a point-wise convolution ($\mathbb{R}^{C} \to \mathbb{R}^{C'}$) that are shared for spatial locations (pixels). It can greatly speed up the inference and reduce model size as well. This type of convolutional operations have been proved very effective and establish a new standard for compact CNNs design. In such a model design, the depth-wise convolution only accounts for 3% of the overall FLOPs, while the point-wise convolution takes up 95% of the FLOPs (Howard et al., 2017). To demonstrate that our method can be well combined with the state-of-the-art CNNs consisting of depth-wise separable convolutions, we compare the regular and the proposed adaptive low-rank factorizations to decompose the point-wise convolutional weight matrix ($W \in R^{C \times C'}$).

---

[2]$(H, W, C)$ stands for the height, width and channel of a receptive field in CNNs, respectively.

Table 1: Performance of MobileNet-CIFAR on CIFAR-10 dataset with different rand-$d$, as a ratio to input channel size ($\frac{d}{n}$). Our adaptive mixture method provides consistent performance gain with negligible FLOPs increase.

|  | Original | 1/4 | 1/4 ours | 1/8 | 1/8 ours | 1/16 | 1/16 ours |
|---|---|---|---|---|---|---|---|
| *Accuracy (%)* | 93.04 | 92.92 | **93.01** | 92.67 | **92.9** | 91.92 | **92.37** |
| *FLOPs (M)* | 44.5 | 27.35 | 27.37 | 18.96 | 18.98 | 14.88 | 14.90 |
| *Param. (M)* | 0.32 | 0.194 | 0.214 | 0.13 | 0.147 | 0.098 | 0.115 |

Table 2: Performance for different networks on ImageNet. With negligible FLOPs increase, adaptive low-rank factorizations outperforms regular ones.

| Network | Top 1 | Params | MACs |
|---|---|---|---|
| ShuffleNet (1.5) | 69.0 | 2.9M | 292M |
| ShuffleNet (x2) | 70.9 | 4.4M | 524M |
| MobileNet | 70.6 | 4.2M | 575M |
| Low-rank MobileNet (0.75) | 68.8 | 2.6M | 209M |
| Adaptive Low-rank MobileNet (0.75) | **70.5** | 2.8M | 209M |
| Low-rank MobileNet | 71.7 | 3.4M | 300M |
| Adaptive Low-rank MobileNet | **73.1** | 3.7M | 300M |

Different from RNNs/LSTM where the input vector $h$ is a vector, $h$ in CNNs is a 3D feature map composed of width, height and channels. Since the feature map can have a large spatial size, we do not use direct projection of $h$ to compute $\pi(h)$, instead we use a global mean pooling to reduce the height and width to 1, i.e. $h_{\text{pool}} = \sum_{ij} h_{ijk}/z$ where $i, j$ sum over all values, in width and height, averaged by the size $z$. By default, we set the number of mixture $K$ to the rank $d$, since $d$ has already been quite small observed from Figure 6. Furthermore, we use the sigmoid activation for computing the mixing weights.

We experiment on both CIFAR-10 (Krizhevsky & Hinton, 2009) and large-scale ImageNet (Deng et al., 2009) datasets. Specifically, we apply the regular and adaptive low-rank factorization on pointwise convolutional kernel $W \in \mathbb{R}^{C \to C'}$ in MobileNet (Howard et al., 2017). For CIFAR experiments, we follow the setting in (Zoph et al., 2017) to build a smaller MobileNet-CIFAR [3] model containing 0.32M parameters and 44.5M FLOPs. For the initial 300 epochs, we train the network with learning rate $0.1$, and halve it for every 25 epochs. For ImageNet experiments, we realize that applying the low-rank factorization for the pointwise convolutional kernel in MobileNet, and adding skip connection, we obtain a network architecture that is the same as MobileNet V2 (Sandler et al., 2018). Therefore, we regard MobileNet V2 as the regular low-rank factorization of the original MobileNet model, and we apply the proposed adaptive low-rank factorization by directly computing mixing weights for the bottleneck layer (illustrated in Figure 3). To make a fair comparison, we follow the same experimental protocol as MobileNet V2 model, including the strategy of learning rate, training epochs, weight decays, etc. (More details can be found in the supplementary materials.)

As a result, Table 1 shows the performance comparison on CIFAR-10 between the regular low-rank factorization and our adaptive mixture method with different rank $d$'s. Our method consistently outperforms the conventional one under different compression ratios. The performance gains are more significant on large compression ratios (or small FLOPs), which demonstrates that our adaptive mixture low-rank factorizations are efficient even for large compression ratios.

For ImageNet, Table 2 shows the comparison of different state-of-art compact convolutional models. We observed that compared to the regular low-rank factorization of MobileNet model (i.e. MobileNet V2), the proposed method achieves significantly better results (2.5% and 2% for two different Low-rank MobileNet settings, respectively), while only adding negligible extra FLOPs (less than 1%).

---

[3]The original MobileNet is built for ImageNet, while CIFAR image has a smaller size $32 \times 32 \times 3$.

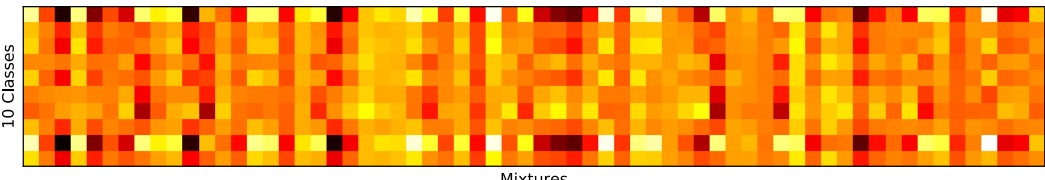

Figure 7: Visualization of the mixtures from the last bottleneck layer in our MobileNet-CIFAR (1/4 size). Each row is averaged from one of the 10 classes. The mixtures show a clear class-discriminative pattern. (Best viewed in color.)

**Visualization of Mixture.** To see whether the mixtures $\pi$ generated for each sample provides class-discriminative information, we visualize the values of mixtures in CIFAR experiments. We record all the mixing weights on CIFAR-10 validation set and average them for each classes. The results are shown in Fig 7, we find the distribution of the mixtures are different for each classes. It shows that the adaptive mixture is able to capture class-discriminative features.

## 5 RELATED WORK

Our work is mostly related to model compression techniques to improve the efficiency of large neural networks. It has been shown that parameters of many modern neural networks are largely redundant (Han et al., 2015a; Ullrich et al., 2017). To reduce the redundancy, various pruning techniques are widely explored (Han et al., 2015b; Ullrich et al., 2017; Louizos et al., 2017). It turns out that more than 90% of connections in a large weight matrix can be pruned without significant loss of information (Han et al., 2015b). This also results in many sparse matrices that require less computation theoretically, however, in practice, specialized hardware (ASIC or FPGA) is required to speed up sparse computations (Han et al., 2016; 2017).

The low-rank factorization of large weight matrices is also a commonly used compression technique (Lu et al., 2016; Nakkiran et al., 2015; Jaderberg et al., 2014; Yu et al., 2017). It does not have the sparse computation issue as in most pruning-base methods. However, the use of a small rank conflicts with the expressiveness as well as the performance of neural networks. Other efforts to improve the efficiency include designing more efficient convolutional operators (Chollet, 2016; Howard et al., 2017; Sandler et al., 2018) and apply quantization on neural network weights (Wu et al., 2018; Li et al., 2016; Achterhold et al., 2018). It is worth noting that the quantization technique is orthogonal to our work and may be integrated together.

The weight matrix in our method is adaptive according to its input. This is also related to dynamic weight generations (Ha et al., 2016; Jia et al., 2016; Hu et al., 2017). The dynamically generated weights can be flexible, but the computational cost for generating the weights during both training and inference can be expensive. In our method, we add the adaptiveness using unnormalized mixture of low-rank factorizations, thus our method can take advantage of dynamic weights while reducing the computation cost.

The use of mixing weights of multiple low-rank factorizations also resembles the mixture of experts (Jacobs et al., 1991; Shazeer et al., 2017; Yang et al., 2018). While both methods use adaptive weights, ours utilizes an unnormalized mixture, and found that normalized mixture can lead to even worse performance. We suspect that unnormalized mixture can better boost the expressiveness for small ranks by more actively using more than one "low-rank factorizations".

## 6 CONCLUSIONS

In this paper, we propose a generic adaptive mixture of low-rank matrix factorization framework, which dynamically incorporate low-rank factorizations with data-dependent weighting based on the input. Our experimental results show that the proposed adaptive mixture method can significantly improve the performance of low-rank factorization on both recurrent and convolutional neural networks. Our method not only keeps the efficiency of low-rank factorization, but also is comparable to (and often outperforms) the accuracy of their full-rank counterparts.

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

## A    MORE DETAILS AND ANALYSIS ON THE PROPOSED MODEL

**Increased expressiveness**    The following proposition demonstrates that the increased expressiveness of the proposed adaptive mixtures: there is no constant weight matrix, even full-rank ones, can mimic the data dependent dynamic weights.

**Proposition 1.** *Given a set of low-rank factorization matrices $\{U^{(k)}\}$ and $\{V^{(k)}\}$ ($U \in \mathbb{R}^{m \times d}$, $V \in \mathbb{R}^{n \times d}$), $W^{(k)} = U^{(k)}(V^{(k)})^\top$, and input data $H = (h_1, h_2, \cdots, h_N)^\top \in \mathbb{R}^{N \times m}$. There exists a data-dependent mixing weight function $\pi(\cdot) : \mathbb{R}^n \to \mathbb{R}^K$, where $\pi(h) = (\pi_1(h), \pi_2(h), \cdots, \pi_K(h))$ and*

$$h_i' = \left( \sum_k \pi_k(h_i) W^{(k)} \right) h_i \text{ and } H' = (h_1', h_2', \cdots, h_N')^\top,$$

*such that for some non-zero $\epsilon$, and any $P \in \mathbb{R}^{m \times n}$, $\|H' - HP\|_F > \epsilon$.*

*Proof sketch.* We can prove it by contradiction. Assuming that $\|H' - HP\|_F = 0$, then there will be a constant matrix $W$ such that $W = \sum_k \pi_k(h_i) W^{(k)}$, which cannot be true since we can freely choose function for computing $\pi_k$.

## B MORE DETAILS AND EXPERIMENTAL RESULTS FOR COMPRESSING RECURRENT NEURAL NETWORKS

### B.1 LONG-SHORT TERM MEMORY AND LANGUAGE MODELING

Language model is a fundamental task in natural language processing. Its goal is to accurately compute the likelihood of a sequence of words, and this can be formulated as follows.

$$P(w_{1...T}) = P(h_0)P(w_1|h_0)P(w_2|w_1, h_1) \cdots P(w_t|w_{T-1}, h_T)$$

where $h_t$ is used to summarizes the context information between the $t$-th word. It is very common to use a recurrent neural network to model the sequence (Zaremba et al., 2014).

Long-short term memory (LSTM) (Hochreiter & Schmidhuber, 1997) is a widely used variant of vanilla recurrent neural networks, where it is shown better to deal with gradient vanishing problem. The LSTM can be summarized as follows.

$$
\begin{aligned}
f_t &= \sigma_g(W_f x_t + U_f h_{t-1} + b_f) \\
i_t &= \sigma_g(W_i x_t + U_i h_{t-1} + b_i) \\
o_t &= \sigma_g(W_o x_t + U_o h_{t-1} + b_o) \\
j_t &= \sigma_c(W_c x_t + U_c h_{t-1} + b_c) \\
c_t &= f_t \circ c_{t-1} + i_t \circ j_t \\
h_t &= o_t \circ \sigma_h(c_t)
\end{aligned}
$$

To simply our implementation, we merge different weight matrices in LSTM above, i.e. $(W_f, U_f, W_i, U_i, W_o, U_o, W_c, U_c)$, into a single matrix below.

$$
Q = \left[ \begin{array}{cccc} W_f & W_i & W_o & W_c \\ U_f & U_i & U_o & U_c \end{array} \right]
$$

We also concatenate the $t$-th hidden state $h_t$ and the $t$-th input vector $x_t$ into a single vector $(x_t^T, h^T)^T$, so that $(f_t^T, i_t^T, o_t^T, j_t^T)^T = Q(x_t^T, h_t^T)^T$. With this re-organization of the computation, we can directly factorize large weight matrix $Q$.

As for the FLOPs, we only count the multiplications and adds of the LSTM layers (excluding the output softmax layer). The non-linear activations are not included. The major extra non-linear activations are in the computation of mixing weights, so the extra cost is expected to be small.

For the MIX-POOL-PJ method, we use segmentation-based pooling for $h$ in computing $\pi$. We make sure that the pooling reduce $h$ to the same dimension as the rank $d$ in our experiments.

### B.2 MORE EXPERIMENTAL RESULTS

In the main paper, we present the trade-off curves between performance/perplexity and computational FLOPS in a LSTM cell. Figure 8 shows similar trade-off curves between performance/perplexity and the number of parameters. While the results are fairly similar to the FLOPS curves, one of the major differences is for the random projection since it has the same FLOPS but no (trainable) parameters compared to the regular projection of input vector $h$. For storage, we can simply save a random seed instead of the whole transformation weight matrix. We find that random projection actually work reasonably well considering it does not containing trainable parameters.

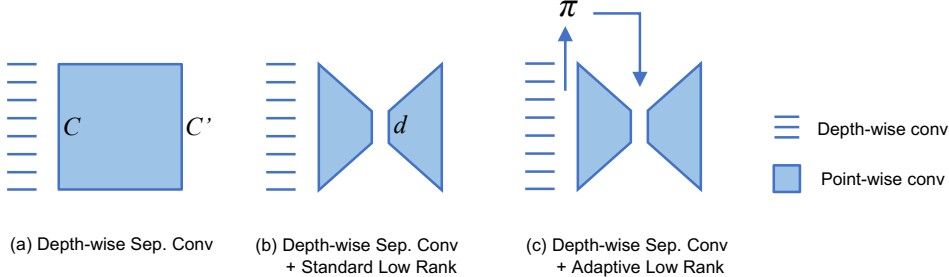

(a) Depth-wise Sep. Conv     (b) Depth-wise Sep. Conv     (c) Depth-wise Sep. Conv
                      + Standard Low Rank         + Adaptive Low Rank

Figure 9: Illustration of low-rank decomposition in depth-wise separable convolutions. No non-linearity is added in the bottleneck.

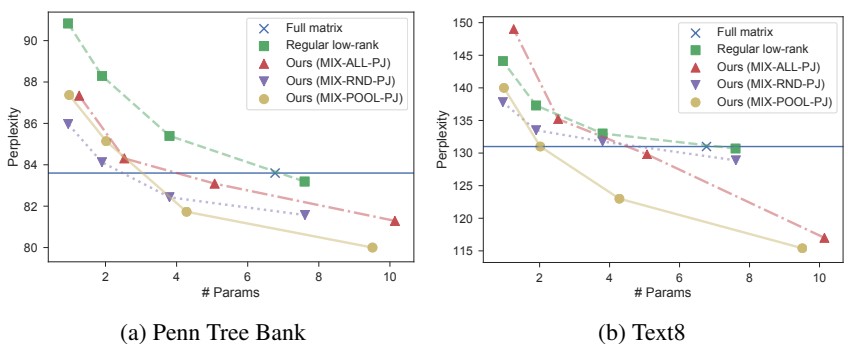

(a) Penn Tree Bank                           (b) Text8

Figure 8: Number of parameters vs perplexity. The horizontal line is for the full matrix baseline, and different color/shape indicating different methods: including the non-adaptive and adaptive low-rank factorizations, with three ways of generating the adaptive maxing weights. The smaller perplexity is more desirable.

## C  MORE DETAILS AND EXPERIMENTAL RESULTS FOR COMPRESSING CONVOLUTIONAL NEURAL NETWORKS

### C.1  LOW-RANK FACTORIZATION FOR POINTWISE CONVOLUTIONAL KERNEL

As mentioned in main paper, we apply the proposed approach for the already compact models with *depth-wise separable convolutions* (Chollet, 2016; Howard et al., 2017; Sandler et al., 2018). That is, we use (adaptive) low-rank factorization to decompose the pointwise convolutional kernel in MobileNet model, and this process can be illustrated in Figure 9.

### C.2  THE NETWORK STRUCTURE FOR MOBILENET-CIFAR

As mentioned in the main paper, we build upon the state-of-the-art compact model, namely MobileNet (Howard et al., 2017). Since the original MobileNet is design for ImageNet and thus too large for images in CIFAR-10, with size of $32 \times 32 \times 3$, we construct a smaller MobileNet-CIFAR architecture. The detailed structure of MobileNet-CIFAR is provided in Table 3. A mobile cell is a depth-wise separable convolution, including a depth-wise $3 \times 3$ convolution and a point-wise $1 \times 1$ convolution. We conduct low-rank factorization on the weight of point-wise convolutions.

### C.3  CONNECTION BETWEEN MOBILENET V1 AND V2 VIA LOW-RANK FACTORIZATION

In our ImageNet experiments, we directly apply the low-rank factorization on the pointwise convolutional kernel. We realize that applying the low-rank factorization for the pointwise convolutional kernel in MobileNet, and adding skip connection, we obtained a network architecture that is the same as MobileNet V2. (See Figure 10 for details.)

Table 3: MobileNet-CIFAR: Each line describes a sequence of 1 or more identical (modulo stride) layers, repeated for $n$ times. All layers in the same sequence have the same number $c$ of output channels. The first layer of each sequence has a stride $s$ and all others use stride 1. A mobile cell consists of a depth-wise convolution and a point-wise $1 \times 1$ convolution. All spatial convolutions use $3 \times 3$ kernels.

| Input | Operator | $c$ | $n$ | $s$ |
|-------|----------|-----|-----|-----|
| $32^2 \times 3$ | conv2d | 32 | 1 | 1 |
| $32^2 \times 32$ | mobile cell | 64 | 3 | 1 |
| $32^2 \times 64$ | mobile cell | 128 | 4 | 2 |
| $16^2 \times 128$ | mobile cell | 256 | 4 | 2 |
| $8^2 \times 256$ | mean pool $8 \times 8$ | - | 1 | - |
| $1 \times 1 \times 256$ | conv2d $1 \times 1$ | 10 | - | - |

Table 4: MobileNet-ImageNet: Each line describes a sequence of 1 or more identical (modulo stride) layers, repeated for $n$ times. All layers in the same sequence have the same number $c$ of output channels. The first layer of each sequence has a stride $s$ and all others use stride 1. A mobile cell consists of a depth-wise convolution and a point-wise $1 \times 1$ convolution. All spatial convolutions use $3 \times 3$ kernels.

| Input | Operator | $c$ | $n$ | $s$ |
|-------|----------|-----|-----|-----|
| $224^2 \times 3$ | conv2d | 32 | 1 | 2 |
| $112^2 \times 32$ | mobile cell | 64 | 1 | 1 |
| $112^2 \times 64$ | mobile cell | 128 | 2 | 2 |
| $56^2 \times 128$ | mobile cell | 256 | 2 | 2 |
| $28^2 \times 256$ | mobile cell | 512 | 6 | 2 |
| $14^2 \times 512$ | mobile cell | 1024 | 2 | 2 |
| $7^2 \times 1024$ | mean pool $7 \times 7$ | - | 1 | - |
| $1^2 \times 1024$ | conv2d $1 \times 1$ | 1000 | - | - |

Table 5: MobileNet-ImageNet-LR: Each line describes a sequence of 1 or more identical (modulo stride) layers, repeated n times. All layers in the same sequence have the same number $c$ of output channels. The first layer of each sequence has a stride $s$ and all others use stride 1. All spatial convolutions use $3 \times 3$ kernels. The expansion factor $t$ is always applied to the input size.

| Input | Operator | $t$ | $c$ | $n$ | $s$ |
|-------|----------|-----|-----|-----|-----|
| $224^2 \times 3$ | conv2d | - | 32 | 1 | 2 |
| $112^2 \times 32$ | bottleneck cell | 1 | 16 | 1 | 1 |
| $112^2 \times 16$ | bottleneck cell | 6 | 24 | 2 | 2 |
| $56^2 \times 24$ | bottleneck cell | 6 | 32 | 3 | 2 |
| $28^2 \times 32$ | bottleneck cell | 6 | 64 | 4 | 2 |
| $28^2 \times 64$ | bottleneck cell | 6 | 96 | 3 | 1 |
| $14^2 \times 96$ | bottleneck cell | 6 | 160 | 3 | 2 |
| $7^2 \times 160$ | bottleneck cell | 6 | 320 | 1 | 1 |
| $7^2 \times 320$ | conv2d $1 \times 1$ | - | 1280 | 1 | 1 |
| $7^2 \times 1280$ | mean pool $7 \times 7$ | - | 1 | - | |
| $1^2 \times k$ | conv2d $1 \times 1$ | - | 1000 | - | - |

Therefore, our method can be directly applied to the original MobileNetV2 to replace the standard low-rank factorization, by computing the adaptive weights dynamically. We evaluated our models

with width multiplier 0.75 and 1.0 with input size $224 \times 224$. The training is conducted under the same protocol as in the original paper.

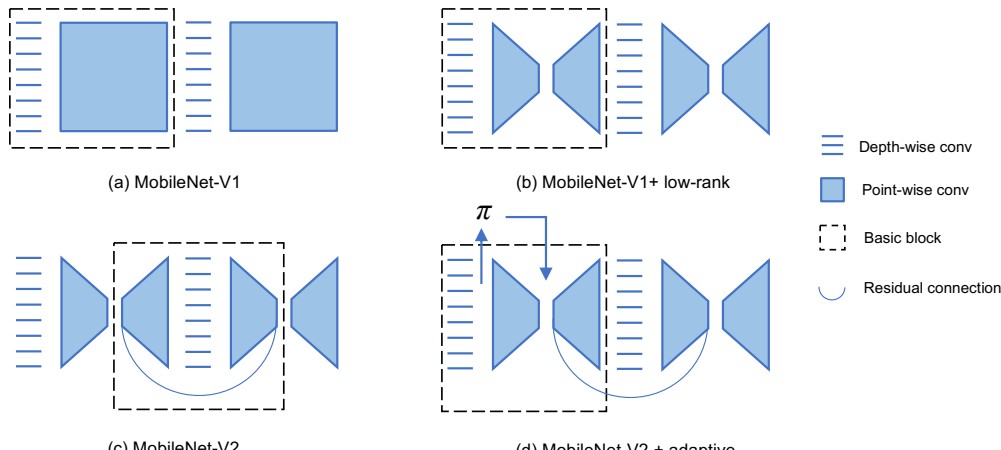

Figure 10: The connection between MobileNet V1 and V2. MobileNet-V2 can be regarded as the low-rank factorized version of V1, where residual connection is added in the bottleneck.

Following (Sandler et al., 2018), no non-linearity other than Batch Normalization is inserted in the bottleneck. The detailed architectures are shown in Table 4 and 5.

## D    EFFICIENT IMPLEMENTATION OF ADAPTIVE LOW-RANK FACTORIZATIONS

Here we introduce a computation technique to avoid splitting a bulk of computation to segments. The default computation with reverse order introduced above can be written as $W(h)h = \sum_{k=1}^{K} \pi_k(h) U^{(k)} \big( \big( V^{(k)} \big)^\top h \big)$, which could introduce $2K$ segmented vector matrix multiplications, and possibly lead to more overheads. However, we could rewrite the same computation as follows.

$$W(h)h = U(\pi(h) \odot (V^T h))$$

Where $\odot$ indicates element-wise multiplication. Since low-rank dimension is usually small, so we (by default) set K to the low-rank dimension $d$, in which case we have $U \in R^{m \times K}$ and $V \in R^{n \times K}$. With this formulation, we do not need to explicit break the computation of regular low-rank of $Wh = UV^T h$ into K branches/mixtures. Rather, we just use non-liear $\pi(h)$ to re-weight the bottleneck vector (i.e. $V^T h$).

## E    TRANSLATING FLOPs INTO WALL-CLOCK TIME

The methods compared in this work are mainly evaluated via accuracy versus FLOPs, as the actual inference time depends on implementation, and also include runtime by other factors (such as final softmax layer in RNN language modeling). In order to access how the FLOPs translate into Wall-Clock Time in a more concentrated environment where we could compare regular low-rank factorization and the proposed counterpart, i.e. adaptive low-rank factorizations.

In this experiment, we measured the actual inference time for (1) linear transformation operator $Wh$, (2) its low-rank factorization $UV^T h$ and (3) the proposed $\sum_{k=1}^{K} \pi_k(h) U^{(k)} \big( V^{(k)} \big)^\top h$. In our experiment, we used pooling before projection to compute $\pi_k(h)$, and the number of mixtures was the same as the low-rank dimension $d$ to enable fast computing. We set both the dimensions of $W$ to 1024, namely $m = n = 1024$. The actual inference time was measured with NVIDIA Jetson TX2 device to simulate edge deployment. Timing was averaged from 500 runs after 200 warm-up runs. The results are shown in Table 6.

Table 6: Inference time / FLOPs of low-rank decomposition.

| Method | low-rank ratio | | | | |
|---|---|---|---|---|---|
| | 1 | 1/2 | 1/4 | 1/8 | 1/16 |
| Full Rank | 68.0ms/1.05M | - | - | - | - |
| Regular LR | - | 68.1ms/1.05M | 34.3ms/0.52M | 17.4ms/0.26M | 8.8ms/0.13M |
| Adaptive LR | - | 84.2ms/1.31M | 39.0ms/0.58M | 18.8ms/0.27M | 9.2ms/0.13M |

We observe that the proposed method provides similar speedup as regular low-rank factorization, especially when the bottleneck layer becomes smaller (which is the case we care more about) and the overhead of computation for mixing weights lessen. With similar speedup, the proposed adaptive low-rank factorization provides better theoretical expressiveness and accuracy/perplexity improvement in practice.

