# OpenReview forum: "Adaptive Mixture of Low-Rank Factorizations for Compact Neural Modeling"
_ICLR.cc/2019/Conference_

### Official Review · AnonReviewer1 · 2018-10-28
**Nice intuitive paper**

**Rating:** 7
**Confidence:** 5

**Review:**

In this paper, the authors propose a compression technique to reduce the number of parameters to learn in a neural network without losing expressiveness.
The paper nicely introduces the problem of lack in espressiveness with low-rank factorizations, a well-known technique to reduce the number of parameters in a network.
The authors propose to use a linear combination of low-rank factorizations with coefficients adaptively computed on data input. Through a nice toy example based on XNOR data, they provide a good proof of concept showing that the accuracy of the proposed technique outperforms the classical low-rank approach.
I enjoyed reading the paper, which gives an intuitive line of reasoning providing also extensive experimental results on multilayer perceptron, convolutional neural networks and recurrent neural networks as well.
The proposal is based on an intuitive line of reasoning with no strong theoretical founding. However, they provide a quick theoretical result in the appendix (Proposition 1) but, I couldn’t understand very well its implications on the expressiveness of proposed method against classical low-rank approach.

---

> ### Author Response · Authors · 2018-11-13
> **Thanks for your feedback!**
>
> We would like to thank the reviewer for the time and valuable feedback. We also like to add on the implication of proposition 1, which demonstrates that the mixture of low-rank factorizations are actually learned non-linear transformation, which is more expressive than the linear one of regular low-rank factorization, i.e. the former cannot be approximated by the latter.

---

### Official Review · AnonReviewer2 · 2018-11-05
**The paper introduces a low rank factorization strategy for neural network compression. They propose a data adaptive model to approximate the weights as a learned mixture of low rank factorizations. The method is novel and the results look promising. A limitation of the method presented is that its is applicable only to weights arising as mode-2 tensors (matrices).**

**Rating:** 6
**Confidence:** 4

**Review:**

Some suggested improvements follow below

1. It is claimed (page 2, last paragraph) that the proposed method leads to a 3.5% and 2.5% improvement in top-1 accuracy over the mobilenet v1 and v2 models. However the results in table 2 indicate 2.5% and 1.4% improvement. This should be corrected.
2. The authors should include the performance of the full rank CNN for the toy example in Figure 1. A Neural Net with 2 neurons in the hidden layer can not learn the XOR/XNOR efficiently . So its rank-1 factorization can only perform as good as the original CNN.
3. In (1), the dimensions of U^k and V^k should be mentioned explicitly.
4. The choice of “k” in (1) should be discussed. How does it relate to the overall accuracy / compression of the CNN?
5. The paper addresses low rank factorization for “MLP”, RNN/LSTM and “pointwise” convolutions. All of these have weights in the form of matrices (mode 2 tensors). The extension to mode-3 and and mode-4 tensors which are more common in CNNs is not straightforward.
6. In the imagenet experiment, the number of mixtures (k) is set to the rank (d). How is the rank computed for every layer?
7. In Fig 7, row 0 and row 8 look identical. Is this indicative of something?

---

> ### Author Response · Authors · 2018-11-13
> **Our response**
>
> We would like to thank the reviewer for the time and valuable comments. Please find our response below.
>
>
> - The results in table 2 indicate 2.5% and 1.4% improvement. This should be corrected.
>
> Thanks for pointing out our typos on the improvement rates. The correct ones should be (1) (70.5-68.8)/68.8=2.5% and (2) (73.1-71.7)/71.7=1.95%. We will update them in the revision.
>
>
> - The authors should include the performance of the full rank CNN for the toy example in Figure 1. A Neural Net with 2 neurons in the hidden layer can not learn the XOR/XNOR efficiently . So its rank-1 factorization can only perform as good as the original CNN.
>
> We’d like to clarify the toy example in Figure 1, the input data point is 2-dimensional, and a MLP (not CNN) is used as function to classify the labels, i.e. P(y|x) = softmax(W’σ(Wx)), where W ∈ R^{2×2}. This is the original full rank model, which is able to effectively learn the synthetic XOR/XNOR task. However, when we factorize W using two 2×1 matrices, i.e. W = UV^T, the induced linear bottleneck largely degenerates the performance (Figure 1b). After applying the proposed method, the performance can be largely improved (Figure 1c).
>
>
> - In (1), the dimensions of U^k and V^k should be mentioned explicitly.
>
> Thanks for the suggestion, we will mention it in the revision.
>
>
> - The choice of “k” in (1) should be discussed. How does it relate to the overall accuracy / compression of the CNN?
>
> The discussion of k is presented in the experiments. We tested different K, and found that using more mixtures generally leads to better results, although the performance starts to plateau when the number of mixtures is large enough. However, to obtain a larger compression rate and speedup, the rank-d we use in the low-rank factorization can be already small, thus the extras of using different number of mixtures may not differ too much. For this reason, we can just set K as rank-d.
>
>
> - The extension to mode-3 and and mode-4 tensors which are more common in CNNs is not straightforward.
>
> Thanks for the questions. We willingly acknowledge that we only considered low-rank factorization of 2d matrices in the scope of this work, which has already found applications in many deep neural network scenarios. For CNNs (which was targeted in this work), we apply our method with widely-used compact depth-separable convolution layers such that it does not require a direct mode-3 tensor factorization.
>
> We also believe it could be straightforward to extend this framework to high-order tensor factorization with minor adjustments. For example, we could apply our method to CP decomposition (https://en.wikipedia.org/wiki/Tensor_rank_decomposition), simply by extending each mixture from two low-rank vector products to three low-rank vector products.
>
>
> - In the imagenet experiment, the number of mixtures (k) is set to the rank (d). How is the rank computed for every layer?
>
> To ensure a fair comparisons with MobielNets, we simply followed the setting in the original MobileNetV2 paper by setting the number of channel of bottleneck to be ⅙ of the output channels.
>
>
> - In Fig 7, row 0 and row 8 look identical. Is this indicative of something?
>
> In Fig 7, the labels for each row are in the same order as in original CIFAR-10 dataset, namely "airplane, automobile, bird, cat, deer, dog, frog, horse, ship, truck". The row 0 and the row 8 correspond to the class of airplane and the class of ship respectively, which suggests the learned mixtures are class discriminative.

---

### Official Review · AnonReviewer4 · 2018-11-11
**Small improvement but breaks a single efficient computation to multiple computation segments (no wall-clock time reported)**

**Rating:** 4
**Confidence:** 5

**Review:**

The paper proposes an input-dependent low rank approximations of weight matrices in neural nets with a goal of accelerating inference. Instead of decomposing a matrix (a layer) to a low rank space like previous work did, the paper proposes to use a linear combination/mixture of multiple low rank spaces. The linear combination is dynamic and data dependent, making it some sense of non-linear combination. The paper is interesting, however,
I doubt its significance at three aspects:
(1) computation efficiency: the primary motivation of this paper is to accelerate inference stage; however, it might not be wise to break computation in a single low-rank space to segments in multiple low-rank spaces. In the original low-rank approximation, only two matrix-vector multiplications are needed, but this paper increases it to 2*K plus some additional overheads. Although the theoretical FLOPs can be cut, but modern hardware runs much faster when a whole bulk of data are computed all together. Because of this, the primary motivation in this paper wasn't successfully supported by wall-clock time;
(2) low-rank approximation: low-rank approximation only makes sense when matrices are known and redundant, otherwise, no approximation target exists (i.e., what matrix is it approximating?). Because of this, low-rank neural nets [1][2] start from trained models, approximate it and fine-tune it, while this method trains from scratch without an approximation target. Although, we can fit the method to approximate trained matrices, then decomposing a matrix to a mixture of low-rank spaces is equivalent to decomposing to one single low-rank space (the only difference is the combination is data dependent). Therefore, I view this paper more in a research line of designing compact neural nets, which brings me to a concern in (3).
(3) efficient architecture design: essentially, the paper proposes a class of compact neural nets, at each layer of which there are K "low-rank" branches with a gating mechanism to select those branches. However, branching and gating is not new [3][4]. I like the results in Table 2, but, to show the efficiency, it should have been compared with more SOTA compact models like CondenseNet.

Clarity:
How FLOPs reduction are exactly calculated? I am not convinced by FLOPs reduction in the LSTM experiments, since in LSTM (especially in language modeling), FLOPs in the output layer are large because of a large vocabulary size (650x10000 in the experiments). However, output layer is not explicitly mentioned in the paper.

Improvement:
(1) Accuracy improvement in Table 1 is not statistically significant, but used more parameters. For example, an improvement of 93.01% over 92.92% is within an effect of training noise;
(2) It is a little hacking to conclude that a random matrix  P_random has a small storage size because we can storage a seed for recovery. When we deploy models across platforms, we cannot guarantee they use the same random generator and has a consistent implementation;
(3) sparse gating of low-rank branches may make this method more computation efficient.

[1] E. L. Denton, W. Zaremba, J. Bruna, Y. LeCun, and R. Fergus. Exploiting linear structure within convolutional networks for efficient evaluation. In Advances in Neural Information Processing Systems (NIPS). 2014.
[2] M. Jaderberg, A. Vedaldi, and A. Zisserman. Speeding up convolutional neural networks with low rank expansions. In Proceedings of the British Machine Vision Conference (BMVC), 2014.
[3] Szegedy, Christian, Wei Liu, Yangqing Jia, Pierre Sermanet, Scott Reed, Dragomir Anguelov, Dumitru Erhan, Vincent Vanhoucke, and Andrew Rabinovich. "Going deeper with convolutions." In Proceedings of the IEEE conference on computer vision and pattern recognition, pp. 1-9. 2015.
[4] Huang, Gao, Danlu Chen, Tianhong Li, Felix Wu, Laurens van der Maaten, and Kilian Q. Weinberger. "Multi-scale dense networks for resource efficient image classification." arXiv preprint arXiv:1703.09844 (2017).

---

> ### Author Response · Authors · 2018-11-13
> **Clarifications: our method does not break a single efficient computation to multiple computation segments**
>
> We thank the reviewer for the time and detailed and valuable comments. Please find our response below.
>
>
> - computation efficiency: in the original low-rank approximation, only two matrix-vector multiplications are needed, but this paper increases it to 2*K plus some additional overheads. Although the theoretical FLOPs can be cut, but modern hardware runs much faster when a whole bulk of data are computed all together.
>
> Thanks for the detailed analysis. We would like to point out that the reviewer’s analysis is based on a specific type of implementation of our method, where K matrix-vector multiplications are conducted. However, this implementation can be easily optimized (especially when we set K to rank where the rank is small): we can still use two matrix-vector multiplications, with a smaller matrix-vector multiplications to compute the mixing weights, and a vector-vector multiplication for weighting bottleneck. This avoids 2*K matrix multiplications. The proposed implementation supports massive parallel computing and also has good data locality, thus it is very efficient to compute.
>
> We willingly admit that the current paper mainly focus the FLOPs, as the actual inference time depends on implementation, and also include runtime by other factors (such as final softmax layer in RNN language modeling). Nevertheless, we conducted some preliminary experiments with non-optimized implementation on RNN (without final softmax), and measure the actual inference time with CPU as shown in the table below. We observed that the proposed method still provides similar speedup as regular low-rank (while getting significantly better accuracy/perplexity).
>
> -----------------------------------------------------------------
>   Method      |      low-rank ratio; time in ms
> -----------------------------------------------------------------
>                       |    1   |  1/2  |  1/4  | 1/8 | 1/16
> -----------------------------------------------------------------
> Full  Rank     | 10.8 | N/A  | N/A | N/A | N/A
> regular LR    | N/A  | 13.1 | 6.6  | 3.3   | 2.0
> adaptive LR  | N/A | 16.5  | 8.6  | 4.5   | 2.8
> -----------------------------------------------------------------
>
> We would like to improve our implementation of the adaptive low-rank to further speed up the actual inference time, and conduct more run-time comparisons in the future.
>
>
> - low-rank approximation: decomposing a matrix to a mixture of low-rank spaces is equivalent to decomposing to one single low-rank space (the only difference is the combination is data dependent).
>
> While we appreciate that the reviewer’s careful observation, especially on that our method can also be applied to the scenarios where the weight matrices are pre-trained. But we would like to point out a inaccurate assertion that our method is equivalent to regular low-rank factorization in [1][2]. Essentially, as shown in proposition 1, our method is non-linear transformation while the regular low-rank is linear. They are not equivalent theoretically, and in practice, as explicitly demonstrated in the toy example (Figure 1), the proposed method is much more expressive than regular low-rank, meaning it can approximate better with around the same amount of parameters and computation. Our experiments on RNN and CNN also show that adaptive low-rank enjoys better expressiveness than conventional low-rank decomposition.
>
>
> - efficient architecture design: branching and gating is not new [3][4]. I like the results in Table 2, but, to show the efficiency, it should have been compared with more SOTA compact models like CondenseNet.
>
> Thanks for the multi-angular analysis. In our understanding, [3] proposes GoogleNet using static branching, and [4] proposes a dynamic network based on input samples. These two papers are orthogonal to our method. We aim to propose an simple yet expressive low-rank decomposition method to speed up the inference of matrix multiplication, which is the fundamental operation in modern neural networks. Therefore, we can apply the method to any existing network architectures.
>
> Regarding empirical evaluation, we aim to show our adaptive low-rank is better than regular low-rank method in our experiments, so we used a very similar yet powerful architecture MobileNet to exclude the interference of other factors, making sure it is a fair comparison. We aim to improve low-rank decomposition itself that is general but not to design a specific compact network architecture.

---

> > ### Author Response · Authors · 2018-11-13
> > **Our clarification (cont'd)**
> >
> > - In LSTM (especially in language modeling), FLOPs in the output layer are large because of a large vocabulary size (650x10000 in the experiments). However, output layer is not explicitly mentioned in the paper.
> >
> > Thanks for point this out. We only calculate the FLOPs of LSTM layer, and do not count the FLOPs in output softmax layer. This is because we only apply the low-rank factorization on the LSTM layers to demonstrate adaptive low-rank enjoys better expressiveness.  We will explicit mention it in the revision. The output softmax is unarguably computation hungry, and can be quite challenging to tackle by itself (there are papers like SVD-softmax trying to tackle this problem), thus we do not include its discussion in this paper.
> >
> >
> > - Accuracy improvement in Table 1 is not statistically significant, but used more parameters.
> >
> > CIFAR dataset is a relatively small and simple dataset, which does not need huge network capacity. Our experiments also show that when the rank is high, both method suffers from little accuracy drop. However, when the rank is low, our method significantly outperforms regular low-rank. Furthermore, the effectiveness of our method is also tested on larger and more convincing dataset ImageNet (in Table 2), where the performance is even more significant.
> >
> >
> > - When we deploy models across platforms, we cannot guarantee they use the same random generator and has a consistent implementation.
> >
> > Yes, we agree the random seed generation can be different across platforms. However, the random projection is just a simple yet interesting comparison for us to better understand the problem. Our main results focus on the one using learned mixing weight computation.
> >
> >
> > - sparse gating of low-rank branches may make this method more computation efficient.
> >
> > Yes, we agree. And we would like to note that our default setting is to set the number of mixtures to the rank so currently it can also be seen as gating, computational wise.

---

> > > ### Comment · AnonReviewer4 · 2018-11-26
> > > **Original rating remains**
> > >
> > > Thanks for feedbacks! I remain my first rating as no new but helpful info provided to solve the concerns, because of following reasons:
> > >
> > > -- Can you clarify what is "weighting bottleneck"? The computation of mixing weight is fine. The explanation regarding "mixing weight" and "weighting bottleneck" is unrelated. Concretely, in Figure 3(b), although one low-rank branch can be done using two matrix-vector multiplications, but there are K branches. The computation segments are also clearly shown in Eq. (1). The "computation efficiency" problem is further supported by the table kindly provided by the author(s):
> > > -----------------------------------------------------------------
> > >   Method      |      low-rank ratio; time in ms
> > > -----------------------------------------------------------------
> > >                       |    1   |  1/2  |  1/4  | 1/8 | 1/16
> > > -----------------------------------------------------------------
> > > Full  Rank     | 10.8 | N/A  | N/A | N/A | N/A
> > > regular LR    | N/A  | 13.1 | 6.6  | 3.3   | 2.0
> > > adaptive LR  | N/A | 16.5  | 8.6  | 4.5   | 2.8
> > > -----------------------------------------------------------------
> > > In regular LR, breaking one matrix to two matrices even increases wall-clock time from 10.8ms to 13.1ms when the ratio is 1/2, proving the disadvantage generated by splitting a bulk of computation to segments; in adaptive LR, breaking it to 2*K consistently increases wall-clock time.
> > > I agree the problem may be alleviated with optimization efforts on library or hardware, then it is unclear how good/worse will it be when compared with fine-grain pruning solutions (Han et al. 2015b & Han et al. 2016), which achieved a higher FLOP reduction.
> > >
> > > -- I agree that mixing coefficients are from non-linear neural networks, while the mixing step is linear once coefficients are known. The only difference is the coefficients are "dynamic and data dependent, making it some sense of non-linear combination".
> > >
> > > -- FLOPs in the output layer are missing when reporting computation reduction. SVD-softmax will deteriorate/compress the model capacity, making compression in LSTMs more challenging.

---

> > > > ### Author Response · Authors · 2018-11-27
> > > > **A simple rewrite of equations can make sure NO splitting a bulk of computation to segments**
> > > >
> > > > Thanks for your follow-up feedback! We would like to provide further clarifications.
> > > >
> > > > -- Can you clarify what is "weighting bottleneck"? Concretely, in Figure 3(b), although one low-rank branch can be done using two matrix-vector multiplications, but there are K branches. The computation segments are also clearly shown in Eq. (1).
> > > >
> > > > The K branches in Figure 3(b) and Eq. (1) are *conceptual* and for the ease of understanding. However, *computational* wise, they can be done with regular big chunk of matrix multiplication. More specifically, linear transform with eq. (1) could be re-written as W(h) h = U (\pi(h) \odot (V^Th)), where \pi(h) weights bottleneck (V^Th) before another linear transformation, instead of \sum_k (\pi(h)_k U^k (V^k)^T h). Since low-rank dimension is usually small, so we (by default) set K to the low-rank dimension, i.e. V \in R^{d\times K}. With this formulation we DO NOT need to break the regular low-rank of Wh = UV^Th into K branches, rather we just use non-linear \pi(h) to re-weight the bottleneck (V^Th). We added a section in last to second section in appendix to describe this process more clearly.
> > > >
> > > > Regarding to the slower wall-clock time for ½ bottleneck, it is actually due to the increased FLOPs, NOT that splitting a bulk of computation to segments (note that both regular and adaptive low-rank have the same issue). For a non-square matrix, introducing ½ bottleneck could actually increase the FLOPs. E.g. splitting 650 x 1500 to 650x538 and 538x1500 (where 538 = average(650, 1500) / 2) will result in 19% increase of FLOPs, which is the case for both regular and adaptive low-rank factorizations. When the matrix to factorize is squared matrix, the FLOPs would not increase for ½ bottleneck. We provide more rigorous experiments for this case in the last section of the appendix.
> > > >
> > > > -- FLOPs in the output layer are missing when reporting computation reduction.
> > > >
> > > > We clarified the FLOPs in the revision to explicit mention it does not include the output layer. The speed-up of softmax (which is itself very challenging) is out of the scope of this work, and our main focus is the main LSTM layers and Convolutional layers. Thus we only apply the low-rank factorization on the LSTM layers to demonstrate adaptive low-rank work better than its regular low-rank counterpart. In the future, we are interested to extend it to broader cases including softmax.

---

> > > > > ### Comment · AnonReviewer4 · 2018-11-27
> > > > > **Computation segments issue is clarified while others remain**
> > > > >
> > > > > Thanks for the positive clarifications, especially appendix D on how to merge computation segments into two matrix-vector multiplications. A add-on, the equation should be able to generalize to K<d, where the mixing coefficients just need to be duplicated and tiled in a typical way.
> > > > >
> > > > > On the speed when rank ratio is 1/2:
> > > > > 1) "low-rank ratio" is not clearly defined. It is misleading to refer to 2d/(m+n) as "low-rank ratio". Conventionally, full rank is min(m, n) and a more reasonable definition of rank ratio is d/min(m, n).
> > > > > 2) the dimension is 1024x1024 square in Table 6, where the FLOPs is the same but wall-clock time increases, validating my previous arguments.
> > > > > 3) the dimension was 650x1500 in a comment but changes to 1024x1024 in the appendix. It may be more reasonable to pick up a dimension used in the benchmarks of CNNs/LSTMs? When converging to a conclusion "With similar speedup, the proposed adaptive lowrank
> > > > > factorization provides better theoretical expressiveness and accuracy/perplexity improvement in practice", the speed should be measured for dimensions in Table 2 for example.
> > > > >
> > > > > Typo:
> > > > > "rand-d" in the caption of table 1.

---

> > > > > > ### Author Response · Authors · 2018-11-28
> > > > > > **Further clarifications on other issues**
> > > > > >
> > > > > > Thanks for the follow-up comments. And we are glad the reviewer now agrees that our method does not break the bulk computation of the low-rank factorization. And that supports what we essentially claimed: the proposed adaptive low-rank method has similar computation cost as regular low-rank factorization, but can improve its theoretical expressiveness and practical accuracy. So for cases where the low-rank factorization can be useful, our method can be seen as a drop-in replacement that improves its performance without extra cost.
> > > > > >
> > > > > > Please find below further clarifications for specific issues mentioned.
> > > > > >
> > > > > > 1) The definition of low-rank ratio, i.e. 2d/(m+n), was given in 3rd paragraph in Sec 4.2 and also mentioned in Figure 6. We agree that d/min(m, n) could be a valid alternative. However, since we use this ratio as a kind of axis and attach FLOPs in every cases, so this does not affect the comparisons of accuracy/perplexity versus FLOPs.
> > > > > >
> > > > > > 2) In table 6, we observe the FLOPs agree pretty well with wall-clock time. For example, 1/16 bottleneck, the FLOPs is almost identical, and the wall-clock time differs within 5% (we didn’t optimize our implementation at kernel level).
> > > > > >
> > > > > > 3) The previous experimental result (in the table of our previous response) misled the reviewer in attributing the increase of time at the 1/2 bottleneck to disadvantage of our method, while our method has similar run-time as regular low-rank. So we chose the squared matrix for experiments to avoid such confusion. However, we will also add the table of non-square matrix result in the revision to relax the choice of target matrix.

---

### Comment · Area_Chair1 · 2018-12-01
**Questions to the authors**

Dear Authors and Reviewers,

Given the disagreements between the reviewers, I took a careful read of the key technical parts of your paper. I have a couple of questions:

1) Is the proposed adaptive mixture of low-rank factorization network trained end-to-end, or trained by approximating Wh, where W is pre-trained?

2) One may consider that using a linear low-rank factorization as a bottleneck layer is not changing the network architecture (in particular, depth of the network, since U(Vh) is just used to approximate Wh for lower computation). In your case, however, I would analogize it to be U*PI(h)*V*h, where the block structured PI(h)=diag[pi(h)_1,...,pi(h)_1,pi(h)_2,...,pi(h)_2,...pi(h)_K] involves a nonlinear transformation of h; from this perspective, your network architecture has actually been modified (i.e., no longer for reducing computation of the same structured network), where a linearly transformed h (i.e. V*h) is reweighed by a nonlinearly transformed h and then linearly transformed by U before sending it to the next layer. The computation saving mainly comes from the fact the unique number of elements in the block structured PI(h) is K. My questions: (1) has anyone tried similar types of architecture but does not impose PI(h) to have a block structure? (2) Is it really that fair to compare the original network with this modified network that effectively has more nonlinear layers? Could you choose deeper networks with similar computation that are trained end-to-end, and compare their performance? Without such type of comparison, I am somewhat hesitate to accept the claims.

3) (As Reviewer 4 also pointed out) Error bars were added to none of the figures and tables, making it difficult to judge whether 1.x% or 2.x% improvement is statistically significant.

Thanks,
AC

---

> ### Author Response · Authors · 2018-12-03
> **Our responses**
>
> Dear AC,
>
> Thanks for your time and detailed comments. Please find our responses below.
>
> - Is the proposed adaptive mixture of low-rank factorization network trained end-to-end, or trained by approximating Wh, where W is pre-trained?
>
> In this work, both low-rank and the proposed method are evaluated based on end-to-end training. However, as an extension to low-rank factorization, our method has arguably the same scope of applicable scenarios as the low-rank factorization, meaning that it could be both applied end-to-end (Linear bottleneck in MobileNet V2) and/or by approximating a pre-trained W.
>
> -has anyone tried similar types of architecture but does not impose PI(h) to have a block structure?
>
> There are some work (Jia et al., 2016; Ha et al., 2016) utilizes non-block-diagonal PI(h) but without U and V, basically, it becomes PI(h)*h, where PI(h) are fully adaptive. However, their methods are not scalable as PI(h) could be high-dimensional. For example, a 3x3 convolutional kernel with 256 channels could have 3*3*256*256 parameters, thus the output dimension of PI(h) function can be ~0.6M. In our method, we utilize the block-diagonal structure to make the original weight matrix W dynamic, i.e. W(h) = U*PI(h)*V, and efficient at the same time. This provides another perspective of viewing our method: enable dynamic weights with efficient computation.
>
> - Is it really that fair to compare the original network with this modified network that effectively has more nonlinear layers?
>
> Our main comparisons are between two methods: (1) regular low-rank factorization, i.e. Wh = U*V*h, and (2) the proposed adaptive mixture of low-rank method, i.e. Wh = \sum_k \pi(k) * U_k*V_k*H = U * PI(h) * V * h. Between these two, the number of layers are the same (although our method turns the linear bottleneck into a essentially non-linear bottleneck). The computation cost (FLOPs) is also very close, and we showed that latter performance significantly better. While it is interesting to compare the proposed method with other compact architectures in the wild (note: we did compare to MobileNet V2 which uses low-rank factorization and was a fairly recent SOTA), or debating whether the low-rank factorization is a good/fair way to factorize deep nets, our contribution mainly focuses on improving the well known low-rank technique with a simple and efficient technique, which also sheds the light to possibly deep nets with scalable dynamic weights as mentioned above.
>
> - Error bars were added to none of the figures and tables, making it difficult to judge whether 1.x% or 2.x% improvement is statistically significant.
>
> We will add standard deviation for Cifar10 in the further revision. For large scale image classification on ImageNet, we would also like to note that most existing literature ignores the error bars (it is a very large dataset with a fixed validation set, and it could be expensive to gather multiple data points that are potentially similar given the training is stable), and >1% is usually considered as significant improvement on this dataset, as evidenced by (Howard et al., 2017, Sandler et al., 2018).
>
> Thanks,
> Authors

---

### Comment · AnonReviewer4 · 2018-12-01
**Missing a list of references on DNN acceleration based on low rank decomposition**

A list of previous publications is missing in the references. Those publications inspired and advanced the low-rank baseline (M. Jaderberg et al 2014) mentioned in this paper. Please consider to cite them.

M. Denil, B. Shakibi, L. Dinh, M. A. Ranzato, and N. de Freitas. Predicting parameters in deep learning. In Advances in Neural Information Processing Systems (NIPS). 2013.

E. L. Denton, W. Zaremba, J. Bruna, Y. LeCun, and R. Fergus. Exploiting linear structure within convolutional networks for efficient evaluation. In Advances in Neural Information Processing Systems (NIPS). 2014.

V. Lebedev, Y. Ganin, M. Rakhuba, I. Oseledets, and V. Lempitsky. Speeding-up convolutional neural networks using fine-tuned cp-decomposition. In International Conference on Learning Representations (ICLR), 2015.

Y. Ioannou, D. P. Robertson, J. Shotton, R. Cipolla, and A. Criminisi. Training cnns with low-rank filters for efficient image classification. In International Conference on Learning Representations (ICLR), 2016.

Y.-D. Kim, E. Park, S. Yoo, T. Choi, L. Yang, and D. Shin. Compression of deep convolutional neural networks for fast and low power mobile applications. In International Conference on Learning Representations (ICLR), 2016.

C. Tai, T. Xiao, X. Wang, and W. E. Convolutional neural networks with low-rank regularization. In International Conference on Learning Representations (ICLR), 2016.

X. Zhang, J. Zou, K. He, and J. Sun. Accelerating very deep convolutional networks for classification and detection. IEEE Transactions on Pattern Analysis and Machine Intelligence, 38(10):1943–1955, Oct 2016.

Wen, Wei, Cong Xu, Chunpeng Wu, Yandan Wang, Yiran Chen, and Hai Li. "Coordinating filters for faster deep neural networks." In The IEEE International Conference on Computer Vision (ICCV). 2017.

---

> ### Author Response · Authors · 2018-12-03
> **Thanks for the suggestions!**
>
> We will review these papers and add them into revision accordingly.

---

### Meta-Review · Area_Chair1 · 2018-12-15
**The paper is clearly written, but there are remaining concerns on contributions and comparisons**

**Confidence:** 4
**Recommendation:** Reject

**Metareview:**

The paper is clearly written and well motivated, but there are remaining concerns on contributions and comparisons.

The paper received mixed initial reviews. After extensive discussions, while the authors successfully clarified several important issues (such as computation efficiency w.r.t splitting) pointed out by Reviewer 4 (an expert in the field), they were not able to convince him/her about the significance of the proposed network compression method.

Reviewer 4 has the following remaining concerns:

1) This is a typical paper showing only FLOPs reduction but with an intent of real-time acceleration. However, wall-clock speedup is different from FLOPs reduction. It may not be beneficial to change the current computing flow optimized in modern software/hardware. This is one of major reasons why the reported wall-clock time even slows down. The problem may be alleviated with optimization efforts on software or hardware, then it is unclear how good/worse will it be when compared with fine-grain pruning solutions (Han et al. 2015b, Han et al. 2016 & Han et al. 2017), which achieved a higher FLOP reduction and a great wall-clock speedup with hardware optimized (using ASIC and FPGA);

2) If it is OK to target on FLOPs reduction (without comparison with fine-grain pruning solutions),
  2.1) In LSTM experiments, the major producer of FLOPs -- the output layer, is excluded and this exclusion was hidden in the first version. Although the author(s) claimed that an output layer could be compressed, it is not shown in the paper. Compressing output layer will reduce model capacity, making other layers more difficult being compressed.
  2.2) In CNN experiments, the improvements of CIFAR-10 is within a random range and not statistically significant. In table 2, "Regular low-rank MobileNet" improves the original MobileNet, showing that the original MobileNet (an arXiv paper) is not well designed. "Adaptive Low-rank MobileNet" improves accuracy upon "Regular low-rank MobileNet", but using 0.3M more parameters. The trade-off is unclear.

In addition to these remaining concerns of Reviewer 4, the AC feels that the paper essentially modifies the original network structure in a very specific way: adding a particular nonlinear layer between two adjacent layers. Thus it seems a little bit unfair to mainly use low-rank factorization (which can be considered as a compression technique that barely changes the network architecture) for comparison. Adding comparisons with fine-grain pruning solutions (Han et al. 2015b, Han et al. 2016 & Han et al. 2017) and a large number of more recent related references inspired by the low-rank baseline (M. Jaderberg et al 2014) , as listed by Reviewer 4, will make the proposed method much more convincing.